# Women's representation as authors of retracted papers in the biomedical sciences

**Ana-Catarina Pinho-Gomes**[1,2]*, **Carinna Hockham**[1], **Mark Woodward**[1,3]

**1** The George Institute for Global Health, Imperial College London, London, United Kingdom, **2** Institute of Health Informatics, University College London, London, United Kingdom, **3** The George Institute for Global Health, University of New South Wales, Sydney, New South Wales, Australia

* a.pinho-gomes@imperial.ac.uk

**Data Availability Statement:** Data cannot be shared publicly because all data were provided by Retraction Watch under a data sharing agreement that does not allow sharing data with third parties.

## Abstract

Women are under-represented among authors of scientific papers. Although the number of retractions has been rising over the past few decades, gender differences among authors of retracted papers remain poorly understood. Therefore, this study investigated gender differences in authorship of retracted papers in biomedical sciences available on Retraction-Watch. Among 35,635 biomedical articles retracted between 1970 and 2022, including 20,849 first authors and 20,413 last authors, women accounted for 27.4% [26.8 to 28.0] of first authors and 23.5% [22.9 to 24.1] of last authors. The lowest representation of women was found for fraud (18.9% [17.1 to 20.9] for first authors and 13.5% [11.9 to 15.1] for last authors) and misconduct (19.5% [17.3 to 21.9] for first authors and 17.8% [15.7 to 20.3] for last authors). Women's representation was the highest for issues related to editors and publishers (35.1% [32.2 to 38.0] for first authors and 24.8% [22.9 to 26.8] for last authors) and errors (29.5% [28.0 to 31.0] for first authors and 22.1% [20.7 to 23.4] for last authors). Most retractions (60.9%) had men as first and last authors. Gender equality could improve research integrity in biomedical sciences.

## Introduction

Peer reviewed papers are retracted when they are considered an invalid source of scientific knowledge and, hence, should be completely removed from the scientific record. According to the Committee on Publication Ethics (COPE) [1], papers should be retracted in the following circumstances: (i) clear evidence for misconduct or honest error; (ii) duplicate publication without proper reference; (iii) plagiarism; and (iv) unethical research. Despite the proposal of the COPE, there is no consensual definition of retraction and the terms retraction, correction, withdrawal, removal, expression of concern, erratum, and corrigendum are often used interchangeably. Although the reasons for retraction are not always clear, they are often provided by retraction notices. According to a previous study, the most common reason for retraction is research misconduct (i.e., fabrication, falsification, or plagiarism), which accounts for about 60% of retractions [2]. Most retractions (about 60%) are initiated by author(s) followed by editors (about 20%), publishers (about 15%), journals (about 5%), and institutions (less than 1%) [3].

Data are available from The Center For Scientific Integrity, the parent nonprofit organization of Retraction Watch, subject to a standard data use agreement. Data are available from team@retractionwatch.com.

**Funding:** The authors received no specific funding for this work.

**Competing interests:** The other authors have no conflicts of interest to declare.

Although retractions are rare, their absolute number and relative proportion of published papers have been increasing over the past two decades [4, 5]. In the biomedical literature, the number and proportion of retractions by year increased between 1980 and 2014 (particularly after 2004) and declined after 2015 [4]. Current estimations for the proportion of retractions are about 4 in 10,000. However, the proportion of retractions varies by publisher, with high impact journals having the highest retraction rates [3]. The cause for this recent rise in retractions is not clear but it seems to be the effect of growing scientific integrity, rather than growing scientific misconduct [6]. Evidence suggests that researchers and journal editors have become more aware of and more proactive about scientific misconduct, without a true increase in cases of fraud.

The underrepresentation of women among authors of scientific papers, particularly in biomedical sciences, has been well documented [7]. Furthermore, the gender gap is larger for last authors than for first authors, thus reflecting the cumulative disadvantage experienced by women in academia and research [8]. Less attention has been dedicated to gender inequalities in retractions of scientific papers. Considering the rise in retractions over the past decades, it is germane to investigate whether gender differences are replicated among authors of retracted papers, overall and for specific reasons [3]. Although gender differences in retractions have been previously identified [9], this study was restricted to papers published in 2016 and included a small sample, which limited its ability to draw meaningful conclusions, particularly regarding reasons for retraction. Therefore, the aim of this study was to reliably estimate the representation of women among authors of retracted papers in biomedical sciences, overall and by reason of retraction, using a large database.

## Methods

We used a database of retracted papers curated by RetractionWatch [10]. This is the largest, most comprehensive database of retracted papers across multiple scientific fields. It was launched in 2010 and is continuously updated by pulling retractions from existing databases, such as PubMed, or publishers' sites. We were provided with a bespoke dataset comprising 35,635 papers in biomedical sciences published between January 1971 and September 2022. No papers were excluded. We extracted data for the first and last author of each paper. We did not include other authors as their contribution and responsibility for the content of papers in biomedical sciences is variable and sometimes minimal. The first author should be the person who contributed most to the work, including writing of the manuscript, whilst the last author is commonly the senior author. We inferred their gender based on their first names using Gender-API software [11]. We accepted gender predictions when the accuracy was estimated to be at least 80% and performed sensitivity analysis for an accuracy of 60% or over. This meant that 14,786 and 15,222 papers were excluded from the main analysis of first and last author, respectively (7,154 and 7,299 papers were excluded from the sensitivity analysis for first and last author, respectively).

We also extracted data for the reasons for retractions, of which there were multiple for each paper. Reasons for retractions were grouped into the following categories (i) misconduct (including misconduct or concerns about authorship), (ii) fraud, (iii) plagiarism, (iv) errors (including errors in any section of the paper), (v) duplication of content (including duplication of data, results, or images), (vi) concerns about data (including results being unreproducible or unreliable, or manipulation of images), (vii) issues related to the journal editor or publisher (including rogue editor, publisher/editor errors, and fake peer review), and (viii) issues related to ethics and law (including non-compliance with ethical standards, lack of approval by ethics, institutional research boards or regulators, lack of patient consent, copyright breaches, and legal threats). Due to small numbers, all remaining reasons were categorised as "other".

We estimated the percentage of retractions that could be attributed to women and men first authors and last authors, overall and for each reason for retraction. We carried out an additional analysis using the gender dyads for first and last authors. We used binomial distributions to estimate percentages retracted by women authors, with 95% confidence intervals, and test whether these percentages were significantly different from 50% (i.e., no gender difference). As a sensitivity analysis, we tested whether women's representation as authors of retractions was comparable to their representation as authors of papers overall. Based on previous literature showing the underrepresentation of women in authorship of biomedical papers, we tested whether those percentages were significantly different from 40% for first authors and 30% for last authors [12, 13].

## Results

Among the 35,635 retracted articles, a total of 20,849 first authors and 20,413 last authors were included in the main analyses after excluding authors whose gender could not be predicted based on first names with at least 80% accuracy. For the sensitivity analyses using a threshold of 60%, a total of 28,481 first authors and 28,336 last authors were included.

Overall, women accounted for 27.4% [26.8 to 28.0] of first authors and 23.5% [22.9 to 24.1] of last authors (**S1 Table**). The sensitivity analysis using a 60% threshold for the accuracy of gender prediction gave similar estimates: women comprised 26.9% [26.3 to 27.4] of first authors and 23.6% [23.2 to 24.1] of last authors. In retractions, there was a steady increase in the percentage of women as first authors from 5.1% in the 1970's to 29.8% in the last decade, whilst the percentage of women as last authors initially decreased and then increased (**Fig 1 and S2 Table**).

Sensitivity analyses comparing women's representation as authors of retractions with their representation as authors of papers overall were comparable to the main analyses. The percentage of women as authors of retractions was significantly lower than their representation as authors of biomedical papers for both first and last author (p<0.0001).

There were differences in women's representation between reasons for retraction (**Fig 2**). For instance, women accounted for 18.9% [17.1 to 20.9] of first authors and 13.5% [11.9 to 15.1] of last authors for fraud but 29.1% [27.5 to 30.8] of first authors and 29.4% [27.8 to 31.1] of last authors for plagiarism. By far the most retractions (60.9%) were authored by men as first and last authors (**Table 1**).

The vertical dashed lines mark the range within the overall representation of women as authors of biomedical science papers has been reported (about 30–40% for first authors and 25–30% for last authors).

## Discussion

Among retracted papers in biomedical sciences, women represented about 27% of first authors and 24% of last authors. These figures reflect an overall increase in women's representation as first and last authors from 1970 to 2022. There were stark differences in reasons for retraction. Whilst women accounted for about 19% of first authors and 14% of last authors for fraud, they comprised about 29% of both first and last authors for plagiarism. Over 60% of retractions had men as first and last authors.

Women's underrepresentation among first and last authors of retracted papers was slightly lower than women's representation among first and last authors of biomedical papers in general. Although there is substantial variation between biomedical fields, women have consistently represented about 30–40% of first authors and 25–30% of last authors [7, 12, 13]. These trends over time are also broadly comparable to general trends in authorship in biomedical

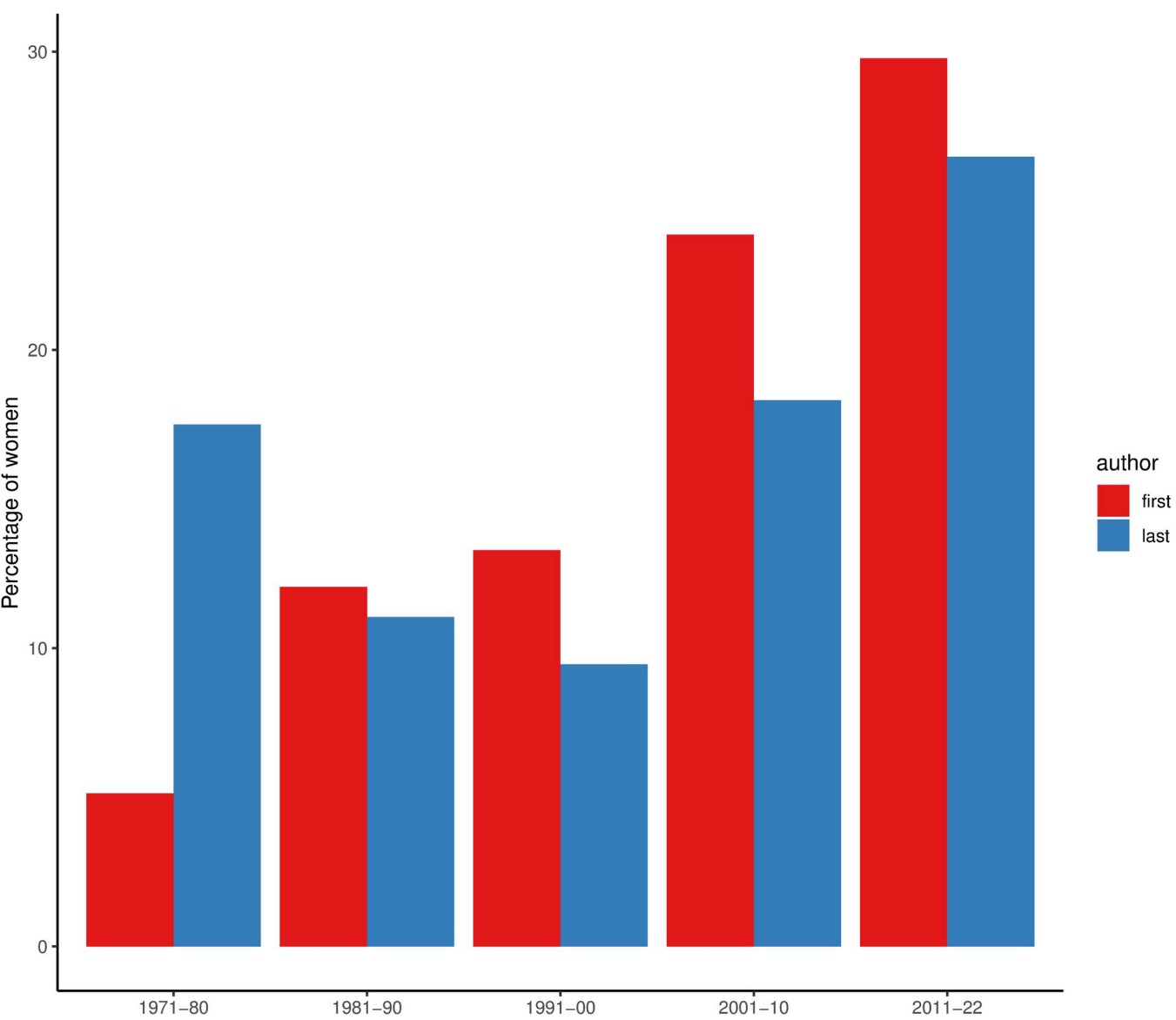

**Fig 1. Women's representation among authors of retracted papers over time.**

sciences [14]. For instance, in medical journals, there was a 4.2% increase in the proportion of women authors between 2008 and 2018, with a larger increase for women as last authors than as first authors (7.8% versus 3.6%, respectively) [12]. We found a greater increase in women as first than last authors, with the percentage of women as first authors remaining larger than last authors over time. This is in keeping with the greater representation of women as first than last authors of biomedical papers overall [12]. The increase in women's representation among authors of retractions is also in line with the rise in the number and proportion of retractions, which seems to be related to greater awareness and commitment to research integrity than an actual increase in fraudulent research [6].

On the other hand, the gender differences in reasons for retraction deserve careful consideration. Women's underrepresentation was greater for misconduct and fraud and lower for plagiarism, duplication of content, and errors. Women's representation was also higher for

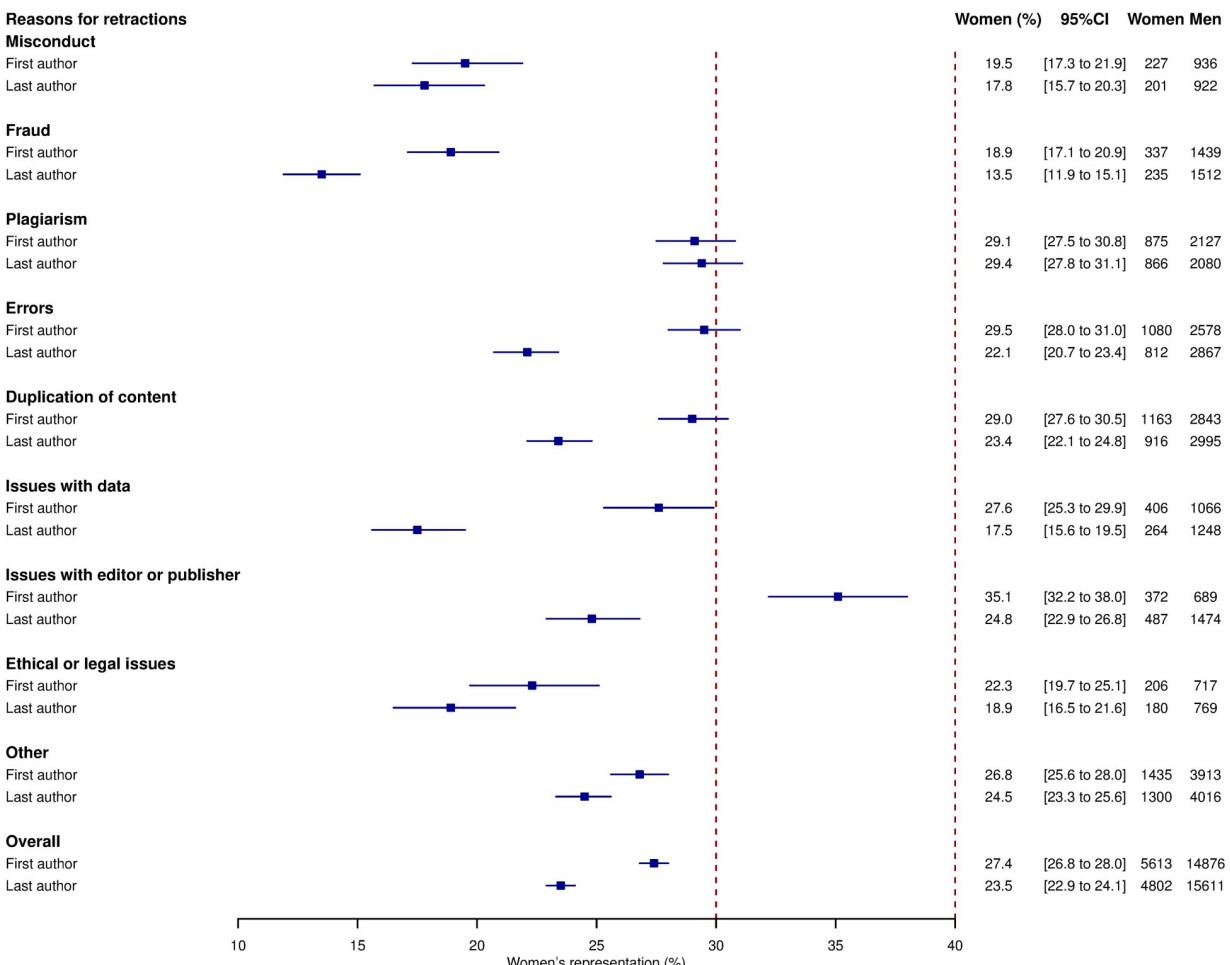

**Fig 2. Women's representation among authors of retracted papers, stratified by reason for retraction.**

issues related to editors and publishers, which are out of author's remit. This is in keeping with previous studies suggesting that men are more likely than women to be involved in fraud and misconduct in research [9, 15]. For instance, in a sample of 113 retractions from diverse scientific fields, fraud and plagiarism accounted for 28.6% of women-authored retractions and

**Table 1. Distribution of authorship of retracted papers by gender dyads, stratified by reason.**

| Reason | Woman-Woman (%) | Woman-Man (%) | Man-Woman (%) | Man-Man (%) |
|---|---|---|---|---|
| Misconduct | 7.4 | 12.8 | 10.0 | 69.8 |
| Fraud | 5.5 | 14.8 | 7.9 | 71.9 |
| Plagiarism | 21.4 | 9.0 | 8.9 | 60.7 |
| Errors | 11.9 | 19.7 | 11.1 | 57.3 |
| Duplication of content | 14.9 | 16.4 | 9.8 | 58.9 |
| Issues with data | 7.6 | 20.8 | 9.9 | 61.7 |
| Issues with editor or publisher | 16.4 | 14.8 | 9.5 | 59.2 |
| Ethical or legal issues | 8.2 | 14.0 | 10.8 | 67.0 |
| Other | 14.7 | 14.3 | 11.3 | 59.7 |
| Overall | 14.2 | 14.7 | 10.3 | 60.9 |

59.2% men-authored retractions [9]. The underlying reasons for these gender differences are difficult to pinpoint. They may be related to attitudes toward research integrity, which are themselves influenced by career goals and ambitions as well as social norms [16, 17]. Gender stereotypes and bias at societal level may result in differences in moral standards and values, which then influence attitudes and behaviours related to research integrity. On the other hand, men may experience greater pressure to maintain a certain level of outputs at senior positions than women [18]. However, no studies have specifically addressed this issue, thus precluding drawing definite conclusions.

Although only a small fraction of biomedical research papers is estimated to be retracted [2], the marked gender differences in underlying reasons for retractions may have important implications. Addressing longstanding gender bias and other barriers that hinder women's progression in academia and research could enhance the integrity and moral standards of the scientific community overall and, hence, reduce misconduct and fraud [19]. Achieving gender equality across the academic ladder, particularly in positions of power and influence, would allow women to serve as role modellers and exert a positive influence on research teams and institutions to adhere to the highest standards of research integrity. This could have far-reaching benefits from increasing public's trust in science [20], to improving the value of research for populations and reducing the long-term harms caused by fraudulent research [21].

Our study is the first to have investigated retractions in relation to gender using a large dataset of biomedical papers published over four decades. However, the large number of papers meant that gender determination had to be based on authors' first names and performed by a software tool. Although this software has been found to be the most reliable at estimating gender, it is not completely accurate, especially for names using non-Latin alphabets [11]. This approach also fails to account for non-binary gender identification as it classifies authors as women or men. Although RetractionWatch is the largest database of retracted papers, it does not capture all retractions as not all publishers clearly label or publicise papers they have retracted or disclose the underlying reasons. In addition, the database was only started in 2010. There is still no consensus about order of authors in biomedical papers, which may have influenced our findings of women's representation for different authorship positions [22]. It is uncertain how generalisable our findings are to other fields beyond biomedical sciences, as practices may vary regarding authorship and retraction criteria between scientific fields.

## Conclusion

Women's representation among authors of retracted papers seems slightly lower than women's representation as authors of biomedical papers overall. Women's underrepresentation is particularly marked for retractions due to fraud and misconduct. Gender equality among authors of biomedical papers could enhance research integrity within the scientific community in general and, hence, reduce the negative impact that retractions have on population health and trust in science.

## Supporting information

**S1 Table. Women's representation among authors of retracted papers.**
(DOCX)

**S2 Table. Women's representation among authors of retracted papers over time (from 1970).**
(DOCX)

## Author Contributions

**Conceptualization:** Ana-Catarina Pinho-Gomes, Carinna Hockham.

**Data curation:** Ana-Catarina Pinho-Gomes.

**Formal analysis:** Ana-Catarina Pinho-Gomes.

**Investigation:** Ana-Catarina Pinho-Gomes.

**Methodology:** Ana-Catarina Pinho-Gomes, Mark Woodward.

**Project administration:** Carinna Hockham.

**Supervision:** Mark Woodward.

**Visualization:** Ana-Catarina Pinho-Gomes.

**Writing – original draft:** Ana-Catarina Pinho-Gomes.

**Writing – review & editing:** Ana-Catarina Pinho-Gomes, Carinna Hockham, Mark Woodward.

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
