## [Decision Letter · Decision Letter 0]

26 Jan 2023

PONE-D-22-33690Women’s representation among authors of retracted papers in biomedical sciencesPLOS ONE

Dear Dr. Pinho-Gomes,

Thank you for submitting your manuscript to PLOS ONE. After careful consideration, we feel that it has merit but does not fully meet PLOS ONE’s publication criteria as it currently stands. Therefore, we invite you to submit a revised version of the manuscript that addresses the points raised during the review process.

The present study provides insights regarding gender differences among retracted papers in biomedical research. It highlights women underrepresentation in scientific publishing consistent with previously published bibliometric analyses. As per PLOSONE main biomedical science target, I consider this topic relevant and in the scope of the journal. Please revise the following issues and submit a revised version of your article at your earliest convenience.

• Reviewers have raised concerns about further contextualization and explanation of study findings, especially how to interpret them considering the findings of previous research. Please address them in the corresponding manuscript sections.

• Please consider further explanation about the systematic selection and adjudication process of included papers to show reproducibility. A flowchart may facilitate readers’ understanding.

• Reviewer #1 has raised a concern about the study's prespecified hypothesis as there is already evidence pointing out women underrepresentation in biomedical sciences. As the main objective of the study is descriptive and exploratory, I consider this issue can be safely removed without compromising study validity. However, existing evidence of this phenomenon should be described in the manuscript.

• As per PLOSONE Data Availability Statement, you must provide all information necessary for interested researchers to apply to gain access to third-party data.

We look forward to receiving your revised manuscript.

Kind regards,

Andres Mauricio Acevedo-Melo, M.D.

Academic Editor

PLOS ONE

Journal Requirements:

"This study was not funded."

"I have read the journal's policy and the authors of this manuscript have no competing interest."

Reviewers' comments:

Reviewer's Responses to Questions

**Comments to the Author**

1. Is the manuscript technically sound, and do the data support the conclusions?

Reviewer #1: No

Reviewer #2: Partly

2. Has the statistical analysis been performed appropriately and rigorously? 

Reviewer #1: No

Reviewer #2: N/A

3. Have the authors made all data underlying the findings in their manuscript fully available?

Reviewer #1: No

Reviewer #2: No

4. Is the manuscript presented in an intelligible fashion and written in standard English?

Reviewer #1: Yes

Reviewer #2: Yes

5. Review Comments to the Author

Reviewer #1: Overall, this is an interesting project, that possess relevant questions, but a larger literature review is needed. The general over or under representation of women in retracted papers can only be accounted if compared with the gender composition among all authors. The differences between the different retraction types are interesting and do not suffer from the problem mentioned above. Nevertheless, much more work on the different rejection types, their motivations and main actors, together with the authorship distribution of tasks and potential responsibilities is needed to make this a valuable contribution. Below some more detailed comments.

Abstract:

The abstract needs some re-writing. An introductory paragraph that contextualizes the research would be helpful. Results should only be a summarized sentence of the abstract, explaining the general conclusions.

Introduction:

The introduction is rather short, and the framing of this project in the previous literature on gender inequalities in science and retracted articles is missing.

Methods

- The data limitations and potential biases should be considered.

- What does each rejection reason imply? Which are the potential motivations between the different rejection types, and how this is appears on the literature? Are different rejection types potentially more related with an authorship position? For example, is there a relation between data manipulation (and rejections related with concerns about data) and first authors? And between issues related with misconduct in authorship or issues with editors and last authors? What does the literature say about this?

- L72 “and test whether these percentages were significantly different from 50%”. This hypothesis is highly problematic. It is well known in the literature that women are underrepresented in science. If the authors want to test gender differences on retraction, they should control by the composition of authors in biomedical sciences between 1971 and 2022 (i.e. the composition at the population level). This might be the most important problem of the article and needs to be address somehow (a limitation statement would not fix this problem). Further controls could be made, for example accounting for the gender composition on the exact set of journals considered, or subfields. If the authors do not have access to a bibliometric database that allows them to compute the gender composition at the authors or paper level, then maybe the hypothesis of underrepresentation of women in the retracted papers should be removed from the analysis. As the literature shows (and the authors noted in the discussion), the proportion of women authors is normally around 30%, which is the same order of magnitude the authors found on retractions. The differences between 27% and 30% can only be significant on comparable datasets.

Results

- The analysis of results needs more work. This is an interesting dataset, but much more work should be done with it. What does the evolution over time say? How does it compare with the gender composition of the field? Which is the relation between first/last author and the different rejection types?

- Also, if the authors have the information, an analysis by disciplines would be interesting.

Reviewer #2: From a bibliometric perspective the paper can be improved by:

- Information on the distribution of the gender of authors in biomedical scientific papers

- Information on the distribution of papers by the number of its authors in biomedical scientific papers

- Discussion on the complexities of author position in a scientific papers and the role of cultural and geographical factors in assigning authors order. For example, Suzetta Burrows & Mary Moore (2011) (Trends in Authorship Order in

Biomedical Research Publications, Journal of Electronic Resources in Medical Libraries, 8: 2,155-168) states that "there are still no universal policies to guide author order in biomedical research publication bylines. Misunderstandings about the placement of a particular author in a sequence of co-authors are common. This, together with changes in indexing policies"

From a sociological perspective the paper can be improved by revising the title as it suggests the study approaches the representation authors of retracted papers have on women: in social sciences and humanities, studies on gender representations follow in-depth and qualitative methodologies. The study follows the participation of women as authors (first and last) of retracted papers in biomedical sciences.

The inclusion of a more descriptive analysis of the gender distribution in the 35.635 papers considered, on the distribution of gender in authorship collaborations in the sample (see for example papers on gender homophile in scientific collaboration).

Authors should address the limitations of the study, a discussion on the generalizability of the empirical results should be included, particularly for strengthening the conclusion that "Gender equality could improve research integrity in biomedical sciences".

6. PLOS authors have the option to publish the peer review history of their article (what does this mean?). If published, this will include your full peer review and any attached files.

Reviewer #1: No

Reviewer #2: No

---

## [Author Response · Author response to Decision Letter 0]

30 Jan 2023

PONE-D-22-33690: Women’s representation as authors of retracted papers in the biomedical sciences 

Answers to reviewers’ comments

Reviewer #1: Overall, this is an interesting project, that possess relevant questions, but a larger literature review is needed. The general over or under representation of women in retracted papers can only be accounted if compared with the gender composition among all authors. The differences between the different retraction types are interesting and do not suffer from the problem mentioned above. Nevertheless, much more work on the different rejection types, their motivations and main actors, together with the authorship distribution of tasks and potential responsibilities is needed to make this a valuable contribution. Below some more detailed comments.

We thank Reviewer #1 very much for their comments.

Abstract:

The abstract needs some re-writing. An introductory paragraph that contextualizes the research would be helpful. Results should only be a summarized sentence of the abstract, explaining the general conclusions.

We added a couple of sentences as background. We did not change the results because the journal allows an abstract of up to 300 words and we considered that it would be useful for readers to have a more detailed description of the results than a single sentence.

Page 2, Abstract

Women are under-represented among authors of scientific papers. Although the number of retractions has been rising over the past few decades, gender differences among authors of retracted papers remain poorly understood. Therefore, this studied investigated gender differences in authorship of retracted papers in biomedical sciences available on RetractionWatch.

Introduction:

The introduction is rather short, and the framing of this project in the previous literature on gender inequalities in science and retracted articles is missing.

We added two paragraphs to the introduction:

Page 3, line 41

Peer reviewed papers are retracted when they are considered an invalid source of scientific knowledge and hence should be completely removed from the scientific record. According to the Committee on Publication Ethics (COPE)(1), papers should be retracted in the following circumstances: (i) clear evidence for misconduct or honest error; (ii) duplicate publication without proper reference; (iii) plagiarism, and (iv) unethical research. Despite the proposal of the COPE, there is no consensual definition of retraction and the terms retraction, correction, withdrawal, removal, expression of concern, erratum, and corrigendum are often used interchangeably. Although the reasons for retraction are not always clear, they are often provided by retraction notices. According to a previous study, the most common reason for retraction is research misconduct (fabrication, falsification, or plagiarism), which accounts for about 60% of retractions.(2) Most retractions (about 60%) are initiated by author(s), followed by editors (about 20%), publishers (about 15%), journals (about 5%), and institutions (less than 1%).(3)

Although retractions are rare, their absolute number and relative proportion of published papers, have been increasing over the past two decades.(4, 5) In the biomedical literature, the number and proportion of retractions by year increased from 1980 to 2014 (particularly after 2004) and declined after 2015.(4) Current estimations for the proportion of retractions are about 4 in 10,000. However, the proportion of retractions varies by publisher, with high impact journals having the highest retraction rates.(3) The cause for this recent rise in retractions is not clear but it seems to be the effect of growing scientific integrity, rather than growing scientific misconduct.(6) Evidence suggests that researchers and journal editors have become more aware of and more proactive about scientific misconduct, without a true increase in cases of fraud.

Methods

- The data limitations and potential biases should be considered.

We expanded the section on the limitations in the discussion as we did not consider it would be appropriate to discuss this in the methods:

Page 7, line 186

Our study is the first to have investigated retractions in relation to gender using a large sample of biomedical papers published over four decades. However, the large number of papers meant that we had to base gender determination by a software tool based on authors’ first names. Although this software has been found to be the most reliable at estimating gender, it is not completely accurate, especially for names using non-Latin alphabets. In addition, this approach fails to account for non-binary gender identification as it classifies authors as women or men.(9) In addition, although RetractionWatch is the largest database of retracted papers, it does not capture all retractions as not all publishers clearly label or publicise papers they have retracted or disclose the underlying reasons.

- What does each rejection reason imply? Which are the potential motivations between the different rejection types, and how this is appears on the literature? Are different rejection types potentially more related with an authorship position? For example, is there a relation between data manipulation (and rejections related with concerns about data) and first authors? And between issues related with misconduct in authorship or issues with editors and last authors? What does the literature say about this?

Although we understand the reviewer’s questions, there is no published literature addressing those questions. We believe the sections added to the introduction (see answer above) summarise the current knowledge about retractions.

- L72 “and test whether these percentages were significantly different from 50%”. This hypothesis is highly problematic. It is well known in the literature that women are underrepresented in science. If the authors want to test gender differences on retraction, they should control by the composition of authors in biomedical sciences between 1971 and 2022 (i.e. the composition at the population level). This might be the most important problem of the article and needs to be address somehow (a limitation statement would not fix this problem). Further controls could be made, for example accounting for the gender composition on the exact set of journals considered, or subfields. If the authors do not have access to a bibliometric database that allows them to compute the gender composition at the authors or paper level, then maybe the hypothesis of underrepresentation of women in the retracted papers should be removed from the analysis. As the literature shows (and the authors noted in the discussion), the proportion of women authors is normally around 30%, which is the same order of magnitude the authors found on retractions. The differences between 27% and 30% can only be significant on comparable datasets.

We added a sensitivity analysis using women’s representation as first and last author of biomedical papers reported in recent literature. We tested whether their representation as authors of retractions was significantly different from their representation as authors of papers overall using a threshold of 40% for first author and 30% for last author. We explained this in the methods and reported the findings in the results:

Page 4, line 109

As a sensitivity analysis, we tested whether women’s representation as authors of retractions was comparable to their representation as authors of papers overall. Based on previous literature showing the underrepresentation of women in authorship of biomedical papers, we also tested whether those percentages were significantly different from 40% for first authors and 30% for last authors.(11, 12)

Page 5, line 127

Sensitivity analyses comparing women’s representation as authors of retractions with their representation as authors of papers overall were comparable to the main analyses. The percentage of women as authors of retractions was significantly lower than their representation as authors of biomedical papers for both first and last author (p<0.0001).

Results

- The analysis of results needs more work. This is an interesting dataset, but much more work should be done with it. What does the evolution over time say? How does it compare with the gender composition of the field? Which is the relation between first/last author and the different rejection types?

We explored our findings in greater detail in the discussion as follows:

Page 5, line 145

Women’s underrepresentation among first and last authors of retracted papers is slightly lower than women’s representation among first and last authors of biomedical papers in general. Although there is substantial variation between biomedical fields, women have consistently represented about 30-40% of first authors and 25-30% of last authors.(7, 11, 12) These trends over time are also broadly comparable to general trends in authorship in biomedical sciences.(13) For instance, in medical journals, there was a 4.2% increase in the proportion of women authors between 2008 and 2018, with a larger increase for women as last authors than as first authors (7.8% versus 3.6%, respectively).(11) We found a greater increase in women as first than last authors, with the percentage of women as first authors remaining larger than last authors over time. This is in keeping with the greater representation of women as first than last authors of biomedical papers overall.(11) The increase in women’s representation as authors of retractions is also in line with the rise in the number and proportion of retractions, which seems to be related to greater awareness and commitment to research integrity than an actual increase in fraudulent research.(6)

- Also, if the authors have the information, an analysis by disciplines would be interesting.

We do not have this information.

Reviewer #2: From a bibliometric perspective the paper can be improved by:

- Information on the distribution of the gender of authors in biomedical scientific papers

We expanded this paragraph in the discussion as follows:

Page 5, line 146

Although there is substantial variation between biomedical fields, women have consistently represented about 30-40% of first authors and 25-30% of last authors.(7, 11, 12) These trends over time are also broadly comparable to general trends in authorship in biomedical sciences.(13) For instance, in medical journals, there was a 4.2% increase in the proportion of women authors between 2008 and 2018, with a larger increase for women as last authors than as first authors (7.8% versus 3.6%, respectively).(11)

- Information on the distribution of papers by the number of its authors in biomedical scientific papers

Please see answer to the comment above. 

- Discussion on the complexities of author position in a scientific papers and the role of cultural and geographical factors in assigning authors order. For example, Suzetta Burrows & Mary Moore (2011), Journal of Electronic Resources in Medical Libraries, 8: 2,155-168) states that "there are still no universal policies to guide author order in biomedical research publication bylines. Misunderstandings about the placement of a particular author in a sequence of co-authors are common. This, together with changes in indexing policies"

This is a valid point, but it is beyond the scope of this paper, which does not aim to explore authorship position but representation of women among authors. We added this to the limitations as follows:

Page 7, line 194

There is still no consensus about the importance of order of authors in biomedical papers and this may have influenced our findings of women’s representation for different authorship positions.(21)

From a sociological perspective the paper can be improved by revising the title as it suggests the study approaches the representation authors of retracted papers have on women: in social sciences and humanities, studies on gender representations follow in-depth and qualitative methodologies. The study follows the participation of women as authors (first and last) of retracted papers in biomedical sciences.

We amended the title as follows:

Women’s representation as authors of retracted papers in the biomedical sciences

The inclusion of a more descriptive analysis of the gender distribution in the 35.635 papers considered, on the distribution of gender in authorship collaborations in the sample (see for example papers on gender homophile in scientific collaboration).

We expanded this section in the discussion as follows:

Page 5, line 145

Women’s underrepresentation among first and last authors of retracted papers is slightly lower than women’s representation among first and last authors of biomedical papers in general. Although there is substantial variation between biomedical fields, women have consistently represented about 30-40% of first authors and 25-30% of last authors.(7, 11, 12) The trends over time are also broadly comparable to general trends in authorship.(13) For instance, in medical journals, there was a 4.2% increase in the proportion of women authors between 2008 and 2018, with a larger increase for women as last authors than as first authors (7.8% versus 3.6%, respectively).(11) We found a greater increase in women as first than last authors, with the percentage of women as first authors remaining larger than last authors over time. This is in keeping with the greater representation of women as first than last authors of biomedical papers overall.(11) The increase in women’s representation as authors of retractions is also in line with the rise in the number and proportion of retractions, which seems to be related to greater awareness and commitment to research integrity than an actual increase in fraudulent research.(6)

Authors should address the limitations of the study, a discussion on the generalizability of the empirical results should be included, particularly for strengthening the conclusion that "Gender equality could improve research integrity in biomedical sciences".

We added a sentence to the limitations as follows:

Page 7, line 196

It is uncertain how generalisable our findings are to other fields beyond biomedical sciences, as practices may vary regarding authorship and retraction criteria between scientific fields.

---

## [Decision Letter · Decision Letter 1]

8 Mar 2023

PONE-D-22-33690R1Women’s representation as authors of retracted papers in the biomedical sciencesPLOS ONE

Dear Dr. Pinho-Gomes,

Thank you for submitting your manuscript to PLOS ONE. After careful consideration, we feel that it has merit but does not fully meet PLOS ONE’s publication criteria as it currently stands. Therefore, we invite you to submit a revised version of the manuscript that addresses the points raised during the review process.

We look forward to receiving your revised manuscript.

Kind regards,

Andres Mauricio Acevedo-Melo, M.D.

Academic Editor

PLOS ONE

Journal Requirements:

Additional Editor Comments:

Dear Authors:

This revised version has addressed the majority of reviewers concerns regarding contextualization of current evidence about gender differences in retracted scientific papers in biomedical sciences. You have also provided information necessary for obtaining access to third-party data according to PLOSONE Data Availability Statement.

I am pointing out minor issues (previous comments included) that require your attention before considering your paper ready for publication:

1. Please consider further explanation about the systematic selection and adjudication process of included papers to show reproducibility: As your study deals with a sample of retracted papers, please provide methods used during paper selection. If not providing a selection flowchart, please provide number of papers per reason of exclusion in your main text.

2. Please address Reviewer #2 concerns about limitations and conclusions based on your findings.

3. Please complete thorough grammar and typos revision on your manuscript.

Reviewers' comments:

Reviewer's Responses to Questions

**Comments to the Author**

1. If the authors have adequately addressed your comments raised in a previous round of review and you feel that this manuscript is now acceptable for publication, you may indicate that here to bypass the “Comments to the Author” section, enter your conflict of interest statement in the “Confidential to Editor” section, and submit your "Accept" recommendation.

Reviewer #1: All comments have been addressed

Reviewer #2: All comments have been addressed

2. Is the manuscript technically sound, and do the data support the conclusions?

Reviewer #1: Yes

Reviewer #2: Partly

3. Has the statistical analysis been performed appropriately and rigorously? 

Reviewer #1: Yes

Reviewer #2: I Don't Know

4. Have the authors made all data underlying the findings in their manuscript fully available?

Reviewer #1: Yes

Reviewer #2: No

5. Is the manuscript presented in an intelligible fashion and written in standard English?

Reviewer #1: Yes

Reviewer #2: Yes

6. Review Comments to the Author

Reviewer #1: I consider that the paper was much improved after the review, and it is ready for publication. As a suggestion, the figure 2 could be improved with the new information of the overall proportion of women as first and last authors in the discipline, maybe with two vertical lines, that would help the reader as a benchmark.

Reviewer #2: The comments have been addressed but some concerns about the paper raised by the reviewers’ comments remain solved. Providing context on the overall distribution of gender among authors of the selected sample would benefit the paper and its argument, as well as more detail on why looking only at the gender of first and last authors makes sense.

In the Abstract, stating that "this studied investigate" makes no sense. The paper needs a thorough revision of grammar and possible typos.

7. PLOS authors have the option to publish the peer review history of their article (what does this mean?). If published, this will include your full peer review and any attached files.

Reviewer #1: No

Reviewer #2: No

---

## [Author Response · Author response to Decision Letter 1]

15 Mar 2023

PONE-D-22-33690: Women’s representation as authors of retracted papers in the biomedical sciences 

Additional Editor Comments:

Dear Authors:

This revised version has addressed the majority of reviewers concerns regarding contextualization of current evidence about gender differences in retracted scientific papers in biomedical sciences. You have also provided information necessary for obtaining access to third-party data according to PLOSONE Data Availability Statement.

I am pointing out minor issues (previous comments included) that require your attention before considering your paper ready for publication:

1. Please consider further explanation about the systematic selection and adjudication process of included papers to show reproducibility: As your study deals with a sample of retracted papers, please provide methods used during paper selection. If not providing a selection flowchart, please provide number of papers per reason of exclusion in your main text.

We were provided with a bespoke dataset of papers retracted in biomedical sciences by Retraction Watch. We added a brief explanation about this source and cited it as recommended by the source. We also included an additional explanation in the limitations. We did not exclude any papers from this dataset. Papers were only excluded due to inability to predict authors based on first names of first or last authors. We clarified this in the manuscript:

Page 4, line 83

We used a database of retracted papers curated by RetractionWatch [10]. This is the largest, most comprehensive database of retracted papers across multiple scientific fields. It was launched in 2010 and is continuously updated by pulling retractions from existing databases, such as PubMed, or publishers’ sites. We were provided with a bespoke dataset comprising 35,635 papers in biomedical sciences published between January 1971 and September 2022. No papers were excluded. We extracted data for the first and last author of each paper. We did not include other authors as their contribution and responsibility for the content of papers in biomedical sciences is variable and sometimes minimal. The first author should be the person who contributed most to the work, including writing of the manuscript, whilst the last author is commonly the senior author. We inferred their gender based on their first names using Gender-API software [11]. We accepted gender predictions when the accuracy was estimated to be at least 80% and performed sensitivity analysis for an accuracy of 60% or over. This meant that 14,786 and 15,222 papers were excluded from the main analysis of first and last author, respectively (7,154 and 7,299 papers were excluded from the sensitivity analysis for first and last author).

Page 10, line 249

In addition, although RetractionWatch is the largest database of retracted papers, it does not capture all retractions as not all publishers clearly label or publicise papers they have retracted or disclose the underlying reasons. In addition, the database was only started in 2010.

2. Please address Reviewer #2 concerns about limitations and conclusions based on your findings.

We have responded to Reviewer #2 concerns.

3. Please complete thorough grammar and typos revision on your manuscript.

We have done this.

 

Reviewers' comments:

Reviewer #1: I consider that the paper was much improved after the review, and it is ready for publication. As a suggestion, the figure 2 could be improved with the new information of the overall proportion of women as first and last authors in the discipline, maybe with two vertical lines, that would help the reader as a benchmark.

We have included those lines as suggested by the reviewer and added an explanation in the figure legend. Please see Figure 2. 

Reviewer #2: The comments have been addressed but some concerns about the paper raised by the reviewers’ comments remain solved. Providing context on the overall distribution of gender among authors of the selected sample would benefit the paper and its argument, as well as more detail on why looking only at the gender of first and last authors makes sense.

We clarified in the methods that this study is not based on a “selected sample”. Rather, it is based on the full database of retracted papers in biomedical sciences provided by Retraction Watch. Studies investigating gender inequalities have typically focused on first and last authors because, in biomedical sciences, these are the guarantors of the paper. The first author is usually the lead author who carried out the research and the last author is usually the most senior author who supervised the research. The contribution and responsibility of middle authors is highly variable and the number of authors differs substantially between papers. Therefore, we considered this analysis would not be relevant for this study and focused on first and last authors. We clarified this in the methods:

Page 4, line 87

We extracted data for the first and last author of each paper. We did not include other authors as their contribution and responsibility for the content of papers in biomedical sciences is variable and sometimes minimal. The first author should be the person who contributed most to the work, including writing of the manuscript, whilst the last author is commonly the senior author.

The large database we used only includes retracted papers, whilst, as far as we can find, there is also no reference to compare women’s representation among authors of retracted papers and papers in general. What we can do is to compare it against the first and last authorship by females in medical journals per se. We have added this as supplementary analysis and added the reference lines to Figure 2 to make even more explicit that women’s representation among authors of retracted papers needs to consider women’s overall representation as authors of biomedical papers. Please see Figure 2. 

In the Abstract, stating that "this studied investigate" makes no sense. The paper needs a thorough revision of grammar and possible typos.

Thank-you for pointing this out. We have thoroughly newly reviewed the manuscript and corrected all typos.

---

## [Editor Report · Decision Letter 2]

30 Mar 2023

Women’s representation as authors of retracted papers in the biomedical sciences

PONE-D-22-33690R2

Dear Dr. Pinho-Gomes,

We’re pleased to inform you that your manuscript has been judged scientifically suitable for publication and will be formally accepted for publication once it meets all outstanding technical requirements.

Kind regards,

Andres Mauricio Acevedo-Melo, M.D.

Academic Editor

PLOS ONE

Additional Editor Comments (optional):

Dear Authors:

The revised version of your article has addressed all peer-review comments and it is now ready for publication. Thanks for your effort, congratulations!
---

## [Editor Report · Acceptance letter]

10 Apr 2023

PONE-D-22-33690R2 

Women’s representation as authors of retracted papers in the biomedical sciences 

Dear Dr. Pinho-Gomes:

I'm pleased to inform you that your manuscript has been deemed suitable for publication in PLOS ONE. Congratulations! Your manuscript is now with our production department. 

Kind regards, 

on behalf of

Dr. Andres Mauricio Acevedo-Melo 

Academic Editor

PLOS ONE